# Hybrid Form of Differential Evolutionary and Gray Wolf Algorithm for Multi-AUV Task Allocation in Target Search

**Ziyun Chen** , **Dengsheng Zhang, Chengxiang Wang and Qixin Sha ***

Faculty of Information Science and Engineering, Ocean University of China, Qingdao 266100, China;
chenziyun@stu.ouc.edu.cn (Z.C.); zds@stu.ouc.edu.cn (D.Z.); wangchengxiang@stu.ouc.edu.cn (C.W.)
\* Correspondence: shaqixin@ouc.edu.cn

**Abstract:** For underwater target exploration, multiple Autonomous Underwater Vehicles (AUVs) have shown significant advantages over single AUVs. Aiming at Multi-AUV task allocation, which is an important issue for collaborative work in underwater environments, this paper proposes a Multi-AUV task allocation method based on the Differential Evolutionary Gray Wolf Optimization (DE-GWO) algorithm. Firstly, the working process of the Multi-AUV system was analyzed, and the allocation model and objective function were established. Then, we combined the advantages of the strong global search capability of the Differential Evolutionary (DE) algorithm and the excellent convergence performance of Gray Wolf Optimization (GWO) to solve the task assignment of the Multi-AUV system. Finally, a reassignment mechanism was used to solve the problem of AUV failures during the task's execution. In the simulation comparison experiments, the DE-GWO, GWO, DE, and Particle Swarm Optimization (PSO) algorithms were carried out for different AUV execution capabilities, respectively.

**Keywords:** multiple Autonomous Underwater Vehicles (multiple AUVs); task allocation; Differential Evolutionary Gray Wolf Optimization (DE-GWO) algorithm





## 1. Introduction

In the face of the complex and changing marine environment, manual exploration of the ocean cannot be accomplished without substantial work and a high degree of danger. AUVs have the characteristics of a wide exploration area, low cost, and high flexibility [1–3]. They can replace manual labor in this complex and dangerous environment, are intelligent and autonomous, are safe, efficiently execute high-risk, long-cycle underwater operations, and have become the key technical equipment for human exploration of the ocean [4,5]. So far, AUVs have had outstanding performance in civil and scientific research fields such as marine data acquisition, undersea terrain exploration, and marine search and rescue, as well as in military missions such as demining and minelaying, military support, and intelligence reconnaissance [6,7].

With the continuous development of Multi-AUV technology, the complex marine environment and variable mission requirements have posed new challenges to Multi-AUV systems [8,9]. When the Multi-AUV system carries out cooperative operation task planning, the problem of Multi-AUV task allocation should be solved firstly [10]. For different operating environments and mission requirements, a reasonable task-allocation algorithm can accurately analyze and process the complicated tasks, so that each AUV in the Multi-AUV system can fully utilize its own resources to efficiently and safely complete the underwater cooperative operation tasks, optimizing system performance [11–13].

Currently, the main task-allocation methods used in Multi-AUV systems are the swarm intelligence optimization algorithm and the Self Organizing Map (SOM) algorithm and market auction algorithm, among which the swarm intelligence optimization algorithm shows better characteristics and is widely used in AUV task allocation [14,15].

Market auction algorithms accomplish task allocation by emulating the auction process of a market economy [16]. Lee et al. put forward a resource-based decentralized auction algorithm that assigns tasks to the robot that can complete the task in the shortest amount of time through reciprocal communication between the auctioneer and the robots. However, this algorithm neglects the auctioneer's interests [17]. Wang and colleagues addressed the problem of Multi-AUV task allocation through the development of a distributed auction algorithm incorporating the interests of the auctioneer and a task reward feedback mechanism [18]. Although the auction algorithm above can improve task allocation, it necessitates high real-time communication for the system [19]. Moreover, if there are multiple pending tasks, the order of task releases impacts the overall completion efficiency [20]. For the task-allocation problem among multiple AUVs, the computational complexity escalates as the number of AUVs increases. In large-scale underwater task-allocation problems, market auction algorithms may not provide efficient solutions within a reasonable time frame [21].

Dong and colleagues utilized a velocity synthesis algorithm and the SOM to address task assignment for a Multi-AUV system in a 3D time-varying current environment. The target point functioned as the input layer, while the output layer consisted of the position of each AUV [22]. Then, Zhu et al. proposed an improved self-organizing algorithm for grid confidence to solve the task-allocation problem for multiple AUV in an obstacle environment [23]. Although the above methods have certain advantages in solving the task-allocation problem, similar to the market auction algorithm, the release order of multiple parallel tasks is a key factor that affects the overall effectiveness of the system [24]. For the threshold method, Huizhen Yang et al. used a dynamic ant colony division of labor model to solve the task-allocation problem for a heterogeneous Multi-AUV system [25]. However, the constraints it considers are simpler and do not take into account the diversity of task types. In addition, Cheng et al. added the effect of the ocean current environment to the SOM network to make the task assignment closer to the real conditions [26]. However, the SOM is not applicable to the task assignment of a Multi-AUV system because, after the SOM training is completed, the map structure will remain unchanged unless it is retrained [27]. In dynamic real-world scenarios that necessitate adaptive, real-time decision-making, the SOM may not be sufficiently flexible to promptly adjust to changes in the task demands, potentially leading to sub-optimal or inefficient task performance [28].

Swarm intelligence algorithms represent a group of nature-inspired computational methods that hold significant potential for AUV target search tasks. Among the most-representative algorithms are the Gray Wolf Optimization (GWO), Differential Evolutionary (DE), and Particle Swarm Optimization (PSO) algorithms. The wolf optimization algorithm has the characteristics of a fast convergence speed and few parameters. GWO was proposed by Mirjalili et al., scholars from Griffith University, Australia, in 2014 [29], which optimizes the search by simulating the process of gray wolf's predatory prey activities, and it is easy to apply [30]. Singh et al. modeled a robot model using an integer-coded wolf pack algorithm, but the proposed assumptions were more idealistic [31]. Cao et al. proposed the application of GWO to the path planning of robots, which can enable robots to plan the optimal path without collision [32]. To improve the accuracy of the gray wolf algorithm during the search, Paul and Kaushik changed the linear convergence factor to nonlinear [33]. Hao et al. also used nonlinear parameters to modulate the parameter $\alpha$ in order to achieve a better balance between exploration and exploitation [34]. All of the above literature has made nonlinear improvements to the convergence factor in the algorithm, but they have not taken into account the lack of search diversity in the later stages of the GWO algorithm, which easily falls into the local optimum [35].

The DE algorithm utilizes biological genetics and natural evolution mechanisms to optimize the search results. Its strength lies in its powerful ability to conduct global searches, resulting in a widespread implementation in various fields. The DE algorithm has the advantage of a strong global search capability and is suitable for parallel computation, but the local search capability is limited [36,37]. Chen et al. verified the advantages of the DE algorithm for solving complex problems in group work, offering the possibility of combin-

ing it with the GWO algorithm [38]. Zheng and colleagues employed the DE algorithm to enhance the GWO algorithm and circumvent the issue of local optimality. Currently, the research on this algorithm is purely theoretical, with no practical applications [39].

As a result, the main contributions of this paper are as follows:

- We propose an architecture for a distributed Multi-AUV task-allocation system. The process of Multi-AUV task allocation takes into account energy consumption, state constraints, and capacity constraints, aiming to establish an appropriate objective function and constraint conditions;
- To address the task allocation in the Multi-AUV system, we introduced the Differential Evolutionary Gray Wolf Optimization (DE-GWO) algorithm. While the GWO algorithm exhibits issues with local optimization, the DE algorithm boasts a powerful global search capability. By combining these two algorithms, we can effectively tackle both local and global optimization challenges in the Multi-AUV task distribution;
- In order to overcome the issue of AUVs failing to execute their missions, our system employs a reassignment strategy. Often, when an AUV breaks down, other algorithms continue to assign tasks to the malfunctioning AUV, resulting in poor mission execution. The implementation of the re-allocation strategy serves as an effective solution to address these failures and enhance the overall robustness of the system.

The rest of this paper is structured as follows. Section 2 describes the Multi-AUV tasking model, including the objective function and constraints. Sections 3 and 4 describe and simulate the DE-GWO algorithm. Finally, Section 5 concludes on the result of the research.

## 2. Mathematical Model

### 2.1. Analysis of AUV Working Process

With more constraints on the Multi-AUV cooperative task problem, it is more demanding in terms of the level of Multi-AUV cooperation and problem-solving complexity. Essentially, this is a combinatorial optimization problem with multiple constraints [40]. The objective of task allocation is to determine the optimal mapping from the task set to the multiple AUVs, minimizing the cost of task completion while adhering to both self-imposed and external constraints within the Multi-AUV system.

In this paper, Figure 1 illustrates the architecture of the distributed task-assignment system for multiple AUVs, which takes into account practical constraints such as the energy limitations and capacity constraints for each AUV.

Assuming that there are $N$ $AUV$s and $M$ tasks in the system, the set of $AUV$s is as follows:

$$AUV = [AUV_1, AUV_2, AUV_3, \ldots, AUV_N] \tag{1}$$

The capabilities of a single AUV are described as follows:

$$AUV_i = [AUV_{State}, AUV_{Position}, AUV_{Ability}, AUV_{Resource}, AUV_{Speed}] \tag{2}$$

In this equation, $AUV_{State}$ represents the current status of the AUV, indicating whether it is in standby mode or actively performing a specific task. $AUV_{Position}$ indicates the current position coordinates. $AUV_{Ability}$ describes the capability to complete a task, with only those equipped with the necessary sensors possessing the ability to carry out the corresponding task. $AUV_{Resource}$ and $AUV_{Speed}$ represent the energy capacity of the AUV and navigation velocity, respectively. The set of *Task*s is as follows:

$$Task = [Task_1, Task_2, Task_3, \ldots, Task_M] \tag{3}$$

The parameters of the tasks are described as follows:

$$Task_i = [Task_{State}, Task_{Position}, Task_{Ability}] \tag{4}$$

In this equation, $Task_{State}$ represents the state of the task, indicating that the task is currently in a completed state or an uncompleted state; $Task_{Position}$ represents the positional coordinates of the task; and $Task_{Ability}$ represents the abilities that the AUV needs to have to complete the task.

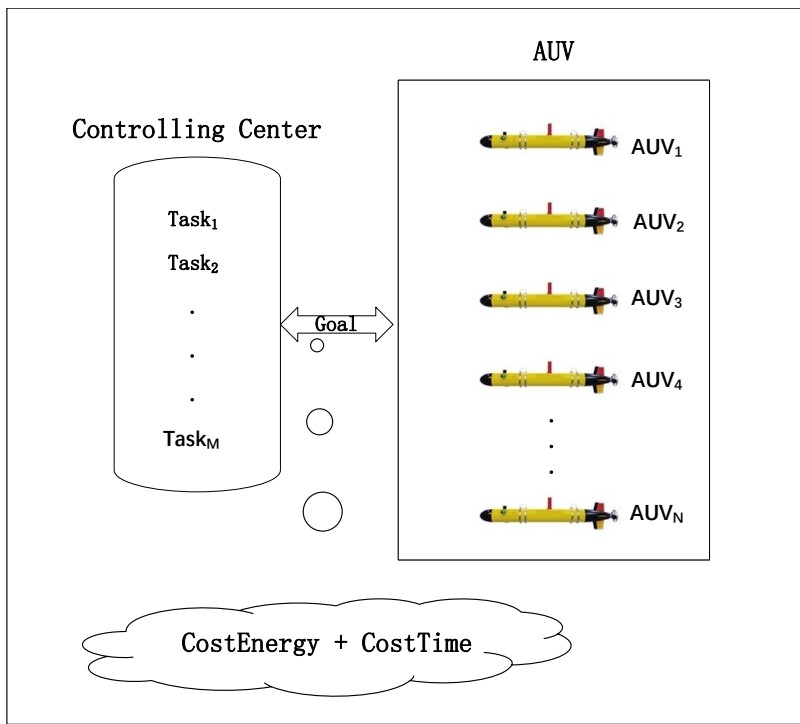

**Figure 1.** Architecture of the proposed distributed task system.

Thus, the Multi-AUV task-assignment problem can be represented as follows:

$$[Task, AUV, Goal] \tag{5}$$

In this equation, *Goal* represents the result of the allocation between *AUV* and *Task*. This results in an optimal task-assignment scheme, ensuring that the assignment outcome is optimal.

### 2.2. Condition Binding

To meet the performance requirements of collaborative task allocation in a Multi-AUV system and to ensure that the allocation of Multi-AUV tasks is both reasonable and effective, the following constraints are considered.

### 2.2.1. State Constraint

Positional constraints are used to ensure that the AUV is in the safe area and to avoid obstacles. The AUV's position $(x_i, y_i)$ must satisfy:

$$g_i(x_i, y_i) \leq 0 \tag{6}$$

In this equation, $g_i$ denotes the inequality condition of the position constraint. Velocity constraints are used to ensure that the velocity of the AUV is in the safe range. This can be expressed as follows:

$$k_i(d_i) \leq 0 \tag{7}$$

In this equation, $k_i$ denotes the inequality condition of the depth constraint.

In general, the number of tasks is larger than the number of AUVs, with each task assigned to a single AUV. To prevent multiple AUVs from executing the same task simulta-

neously and ensure the uniqueness of task execution, state constraints are imposed. The constraint function is defined as follows:

$$\sum_{i=1}^{N} X_{ij} = 1, \forall Task_j \in Task \tag{8}$$

$$X_{ij} \in 0, 1 \tag{9}$$

where $X_{ij} = 1$ means that $AUV_i$ executes the target task $Task_j$ and $X_{ij} = 0$ means that $AUV_i$ does not execute the task $Task_j$.

### 2.2.2. Energy Constraint

Typically, the energy consumed by an AUV is related to the distance of the path it navigates while performing a mission. The distance between the current position of the $AUV_i$ and the $Task_j$ can be expressed as follows:

$$l_{i,j} = \sqrt{(x_i - x_j)^2 + (y_i - y_j)^2} \tag{10}$$

In this equation, $X_i$ and $Y_i$ represent the horizontal and vertical position of the $AUV_i$, while $X_j$ and $Y_j$ represent those of the mission target position. The distance between $AUV_i$ and the mission target point is denoted as $l_{i,j}$. Assuming the distance traveled by $AUV_i$ to perform the task at this time is $dist_i$, the expression of its energy requirements is as follows:

$$E\_cost_i = dist_i \times e \tag{11}$$

In this equation, $e$ represents the energy consumption coefficient, which signifies the energy consumed per unit distance traveled by an AUV. Each AUV has a limited amount of energy due to its load limitations. During task allocation, it is crucial to guarantee that each AUV is equipped with sufficient energy to complete its assigned tasks, while also ensuring that all AUVs collectively possess enough energy to complete all tasks. The energy constraint is expressed as follows:

$$\sum_{i=1}^{N} (E\_cost_i \leq E_i) \tag{12}$$

In this equation, $N$ is the number of AUVs, $E\_cost_i$ represents the energy consumed by $AUV_i$ to execute the sequence of tasks allotted to it, and $E_i$ represents the energy capacity of $AUV_i$.

### 2.2.3. Capacity Constraint

In this paper, tasks are classified into three distinct types based on their specific mission requirements:
1. Missions requiring underwater cameras: These tasks necessitate AUVs equipped with underwater cameras for their successful execution.
2. Missions requiring side-scan sonar: AUVs with side-scan sonar capabilities are essential for these missions.
3. Missions requiring both underwater cameras and side-scan sonar: Some missions require a combination of both underwater cameras and side-scan sonar. AUVs with this combined capability can effectively perform all three types of tasks.

### 2.3. Objective Function

The task of allocating multiple AUVs can be formulated as a multi-objective optimization problem. The evaluation metrics in this optimization problem include *CostEnergy* and

*CostTime*. The goal is to minimize the evaluation function to enhance the overall utility of the Multi-AUV system.

*CostEnergy* is defined as the total energy consumed by all AUVs during the mission. Minimizing energy consumption leads to more-efficient energy utilization and shorter travel distances for the multiple AUVs, ultimately improving mission completion. The *CostEnergy* function is represented as follows:

$$CostEnergy = \sum_{i=1}^{N} \sum_{j=1}^{M} E\_cost_{i,j} \tag{13}$$

*CostTime* represents the time taken by all AUVs to complete their tasks, defined as the maximum time among all AUVs to finish their tasks. The time required for each AUV to complete its task sequence comprises both the navigation time and the task execution time. The voyage time is calculated as the total distance traveled by an AUV divided by its average speed between mission objectives. The mission execution time refers to the duration an AUV spends performing its assigned tasks. The cost of the mission time is expressed as follows:

$$CostTime = max(\sum_{j=1}^{M}(\frac{dis_{i,j}}{v} + Time_j)) \tag{14}$$

In this equation, $dis_{ij}$ represents the distance between $AUV_i$ and the $Task_j$'s point, $v$ represents the navigation speed of the AUV, and $Time_j$ is the time consumed by the AUV to perform $Task_j$. A lower *CostTime* indicates a shorter time for the Multi-AUV system to accomplish the entire task, demonstrating that the task assignment output maximizes the resources of the Multi-AUV system and reduces task execution time. Thus, the objective function is as follows:

$$minJ = CostEnergy + CostTime \tag{15}$$

To fully utilize the effectiveness of the Multi-AUV system under multiple constraints, the principles governing task assignments can be summarized as follows:

1. Capability matching: The Multi-AUV system splits complex tasks into smaller, manageable sub-tasks. These sub-tasks are then assigned to AUVs based on their capabilities and features. For example, if a task requires underwater imaging, it should be assigned to AUVs equipped with appropriate cameras or sensors. This principle ensures that tasks are assigned to AUVs that are best suited for their successful execution, maximizing the efficiency of the system;

2. Minimizing system cost and energy constraint: Task allocation in a Multi-AUV system aims to achieve successful task completion while minimizing the overall cost to the system. This includes considerations such as minimizing fuel consumption, reducing wear and tear on the AUVs, and avoiding unnecessary duplication of efforts. It is critical to ensure that the energy constraint of the AUV system is satisfied, as excessive energy consumption can limit the system's operational range and duration. Effective task allocation can help achieve this balance;

3. Load balancing: Load balancing is a crucial aspect of task allocation in a Multi-AUV system. It involves distributing tasks as evenly as possible among the available AUVs. An even distribution prevents overburdening certain AUVs while leaving others underutilized. This not only maximizes the potential of each AUV, but also helps in achieving efficient task completion. Load balancing may consider factors such as the capacity of the AUVs, their proximity to tasks, and their current energy levels to ensure tasks are allocated in a balanced and fair manner.

## 3. Innovative DE-GWO Algorithm

In order to solve the issue of local optimality during the task allocation process, we propose the DE-GWO algorithm. Additionally, we propose the reassignment strategy for the AUV experiencing faults during task execution. Our algorithm is superior to other methods.

### 3.1. Overview of the Proposed Algorithm

Given the discrete nature of the Multi-AUV task-allocation problem, the GWO algorithm requires a discretization operation to map the task-allocation problem model. The number of tasks to be executed by the Multi-AUV setup determines the location dimension for each gray wolf in the population. For example, if there are N tasks that need completion, each gray wolf's location is represented as an N-dimensional vector. Once the gray wolf population is positioned, we sort each individual's position within the N-dimensional space by magnitude to generate the task sequence for the AUVs. Furthermore, the task sequences are randomly grouped based on the number of AUVs. This random grouping adds an element of discretization to both the algorithm and the task sequences for each AUV. Consequently, the position of each gray wolf signifies how the AUV task sequences are allocated and organized. If the generated task allocation plan fails to meet the Multi-AUV task allocation constraints, we regenerate the locations and groupings of gray wolves until the plan aligns with the task requirements. Through continuous iterations of the algorithm, we ultimately obtain an allocation that optimizes the objective function.

The GWO algorithm faces challenges related to poor population diversity and premature convergence when solving complex problems. To address these issues in the context of Multi-AUV task allocation, this study initially employed Singer chaotic mapping to initialize the gray wolf population, thereby enhancing population diversity. This approach ensures a uniform distribution of gray wolves within the search space, which, in turn, boosts the algorithm's global search capability and introduces greater randomness and diversity into the sequence of Multi-AUV tasks. Singer chaotic mapping is a type of chaotic mapping known for its numerous advantages, including a simple structure, high randomness, and a uniform distribution [41]. Its mathematical expression is as follows:

$$d_{k+1} = \mu(7.86d_k - 23.31d_k^2 + 28.75d_k^3 - 13.302875d_k^4)$$ (16)

In this equation, $\mu \in [0.9, 1.08]$ and $d_k \in [0, 1]$ are used to initialize the gray wolf population position. $X_k$ is defined as follows:

$$X_k = u_{min} + d_k(u_{max} - u_{min})$$ (17)

In this equation, $u_{max}$ and $u_{min}$ are the upper and lower bounds of the search space, respectively.

Meanwhile, this paper employed a strategy of nonlinear convergence to enhance the convergence factor and optimize the position update formula in the GWO algorithm, aiming to boost its global search capabilities. Considering the fast convergence speed of the DE algorithm and its role in strengthening information exchange between individuals in the population while improving population diversity through mutation, this paper introduced the crossover, selection, and mutation operations of the DE algorithm during the evolution of the GWO algorithm. This promotes continuous evolution among gray wolf individuals through the survival of the fittest, thus enhancing the population diversity of the GWO algorithm and improving its ability to escape local optima.

In the standard GWO algorithm, the vector $A$ represents the extent of prey search undertaken by the gray wolf population. The convergence factor $a$ is crucial in balancing the algorithm's local and global search capabilities. The convergence factor $a$ affects the coefficient vector $A$ as follows:

$$a = 2 \times (1 - \frac{t}{T})$$ (18)



$$A = 2a \cdot r_1 - a \tag{19}$$

In this equation, $T$ is the maximum number of iterations of the algorithm and $r_1$ is a random vector between [0, 1]. When solving complex problems, it is important to consider the update strategy of a linearly decreasing convergence factor. The strategy causes the algorithm to fall into local optimum during the iteration. To enhance the algorithm's global search ability, this paper used a nonlinear update strategy for the convergence factor, formulated as follows:

$$a = 2 \cdot (1 - \frac{(e^{\frac{Iter}{MaxIter}} - 1)}{e - 1})) \tag{20}$$

In this equation, *Iter* represents the current number of iterations and *MaxIter* represents the maximum number of iterations of the algorithm. The convergence factor a decreases nonlinearly from 2 to 0 during the iteration process. It takes a larger value in the early iterations, decaying more slowly, thus improving the global search capability. In the late iterations, the convergence factor quickly reduces to 0, facilitating a swift convergence of the algorithm.

In the standard GWO algorithm, $\alpha$ $\beta$ and $\delta$ together guide the gray wolves in the population to update their positions so as to gradually approach the target, and the formula is expressed as follows:

$$\begin{aligned} D_\alpha &= |C_1 \cdot X_\alpha(t) - X_k(t)| \\ D_\beta &= |C_2 \cdot X_\beta(t) - X_k(t)| \\ D_\delta &= |C_3 \cdot X_\delta(t) - X_k(t)| \end{aligned} \tag{21}$$

$$\begin{aligned} X_1(t+1) &= X_\alpha(t) - A_1 \cdot D_\alpha \\ X_2(t+1) &= X_\beta(t) - A_2 \cdot D_\beta \\ X_3(t+1) &= X_\delta(t) - A_3 \cdot D_\delta \end{aligned} \tag{22}$$

$$X_k(t+1) = \frac{(X_1(t+1) + X_2(t+1) + X_3(t+1))}{3} \tag{23}$$

In the aforementioned equation, $X_\alpha(t)$, $X_\beta(t)$, and $X_\delta(t)$ represent the positions of the $\alpha$-wolf, $\beta$-wolf, and $\delta$-wolf, correspondingly, in the present iteration. Additionally, the search steps conducted in adherence to the guidelines of the $\alpha$-wolf, $\beta$-wolf, and $\delta$-wolf are identified by $D_\alpha$, $D_\beta$, and $D_\delta$, respectively. $X_i(t+1), (i = 1, 2, 3)$ represent the positions of individual gray wolves in the population after position updating led by the $\alpha$-wolf, $\beta$-wolf, and $\delta$-wolf, respectively; $A_1$, $A_2$, $A_3$ and $C_1$, $C_2$, $C_3$ are the corresponding coefficient vectors. In order to improve the GWO algorithm's ability to escape local optima and prevent algorithm stagnation when the $\alpha$-wolf, $\beta$-wolf, and $\delta$-wolf are trapped in local optima, we propose a dynamic weighting strategy in this paper. The population's position is updated using the following formula:

$$\begin{aligned} w_i &= \frac{f(X_i)}{f(X_1) + f(X_3) + f(X_2)} \\ X_k(t+1) &= \frac{\sum_{i=1}^{3} w_i \times X_i(t+1)}{3} \end{aligned} \tag{24}$$

In the equation above, the objective function calculates the adaptation values of the $\alpha$-wolf, $\beta$-wolf, and $\delta$-wolf, denoted as $f(X_i)$ for $(i = 1, 2, 3)$, respectively. The corresponding weights of the adaptation values are given by $w_i$ for $(i = 1, 2, 3)$. Following the movement of the gray wolf, the task sequence uses a grouping method that selects the most-effective allocation scheme between the original method and the $\alpha$-wolf grouping method to improve the algorithm's ability to find the optimal solution.

The paper addressed the issue of premature convergence and limited global search capabilities in the standard GWO algorithm. To overcome these challenges, the paper

suggested the DE algorithm, which provides faster convergence and enhances population diversity through information exchange during the mutation process among individuals. Through a comprehensive comparison and analysis of the advantages and disadvantages of both algorithms, this paper introduced the DE-GWO algorithm as an effective solution for Multi-AUV task allocation. When updating the positions of individual gray wolves, the DE algorithm utilizes the mutation, crossover, and selection operations to increase the diversity of the gray wolf population, improve the algorithm's global search capabilities, aid in escaping local optima, and expedite the convergence speed.

After revising the position under the guidance of the $\alpha$-, $\beta$-, and $\delta$-wolves, the individuals of the gray wolf population underwent a mutation operation utilizing the DE/best/1 mutation strategy outlined in Table 1. The mutation process involves selecting the $\alpha$-wolf of the gray wolf population and summing the deviation vectors of two chosen gray wolf individuals after weighting them accordingly, resulting in the creation of a mutated individual. The variant individuals will generate task sequences randomly to group them, which will increase diversity in the allocation scheme. Then, either the variant individual or target individual will be chosen randomly as the newborn individual for the current iteration, as described below:

$$v_{i,j}(g) = \begin{cases} h_{i,j}(g), rand(0,1) \leq CR, randi(1,D) = j \\ X_{i,j}(g), otherwise \end{cases} \tag{25}$$

In this equation, $rand(0,1)$ represents a random number between $[0,1]$, $CR$ denotes the crossover probability, and $h_{i,j}(g)$ signifies the variant individual.

**Table 1.** Differential evolutionary algorithm mutation strategies.

| Variation Strategy | Concrete Form |
|---|---|
| DE/rand/1 | $V_i(g) = X_{p1}(g) + F \cdot (X_{p2}(g) - X_{p3}(g))$ |
| DE/best/1 | $V_i(g) = X_{best}(g) + F \cdot (X_{p1}(g) - X_{p2}(g))$ |
| DE/randtobest/1 | $V_i(g) = X_i(g) + F \cdot (X_{best}(g) - X_i(g)) + F \cdot (X_{p1}(g) - X_{p2}(g))$ |
| DE/rand/2 | $V_i(g) = X_{p1}(g) + F \cdot (X_{p2}(g) - X_{p3}(g)) + F \cdot (X_{p4}(g) - X_{p5}(g))$ |
| DE/best/2 | $V_i(g) = X_{best}(g) + F \cdot (X_{p1}(g) - X_{p2}(g)) + F \cdot (X_{p3}(g) - X_{p4}(g))$ |

$p1, p2, p3, p4, p5$ denote the random individuals in the population, and $p1 \neq p2 \neq p3 \neq p4$, respectively; $F$ is the variation operator with the ability to balance the global search with the local search; $X_{best}(g)$ is the optimal individual in the population in the gth generation; $x_{pi}(g) - x_{pj}(g)$ represent the deviation vectors of the two random individuals; $V_i(g)$ represents the gth generation of the variation vector.

## 3.2. Steps of DE-GWO

The DE-GWO algorithm utilized in this study was derived from the GWO algorithm, which draws parallels between the solution process of the Multi-AUV multi-task issue and the gray wolf searching for prey in the search space. The gray wolf stands for a set of feasible solutions for the Multi-AUV task assignment, with the optimal solution being the location of the prey. The flowchart of the DE-GWO algorithm is shown in Figure 2:

Step 1: First, the necessary algorithm parameters are initialized, including a population $N$ of 20, a $T_{max}$ of 1000, and a $CR$ of 0.8.

Step 2: The population positions are initialized using Singer chaotic mapping and denoted as $X_i = [x_{i1}, x_{i2}, x_{i3}, \ldots, x_{iD}]$. Subsequently, these positions are sorted in ascending order, and the task sequences for the Multi-AUV system are generated. These task sequences are then grouped together to establish the Multi-AUV task allocation scheme.

Step 3: Calculate the objective function value of the individual gray wolves in the population, and rank the individuals based on the size of the objective function value, then select the optimal top three gray wolf individuals denoted as $X_\alpha$, $X_\beta$, and $X_\delta$, respectively.

Step 4: Calculate the convergence factor $a$ using Equations (16) and (17), and determine the values of $A$ and $C$. Update the positions of the individual gray wolves in the population using Equation (22), and compute the updated population's objective function value.

Step 5: Choose the DE/Best/1 mutation strategy from Table 1 for the mutation operation. Then, perform the crossover operation, using Equation (23) to generate a temporary population, and calculate the objective function value for each gray wolf in this population.

Step 6: Perform a one-to-one selection operation between individuals in the temporary population and the individuals in the original population. Retain the individuals with better fitness values in the original population for the next iteration. This selection process ensures that the population evolves over the iterations, with individuals having better fitness values being preserved for further exploration in the search space.

Step 7: Sort the gray wolf individuals in the resulting population after the update according to the size of the fitness value, and update the positions of the gray wolf individuals $X_\alpha$, $X_\beta$, and $X_\delta$, with the top three fitness values.

Step 8: If the current number of iterations reaches the maximum number of iterations $T_{max}$ of the algorithm, the optimal result $X_\alpha$ is output and the algorithm is terminated; otherwise, skip to Step 4.

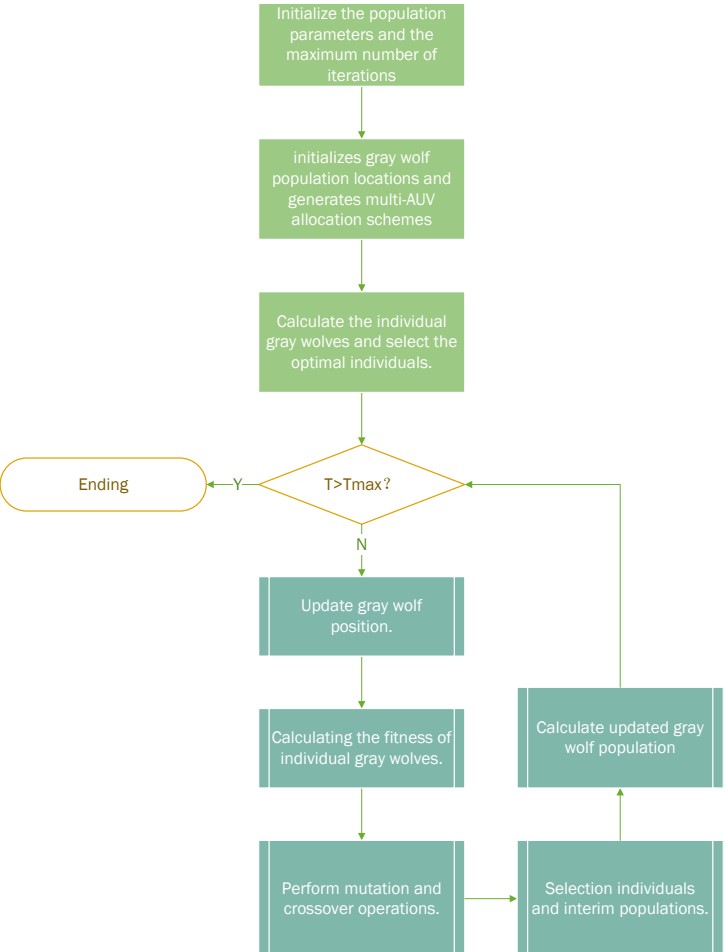

**Figure 2.** Flowchart of DE-GWO.

### 3.3. Mandate Redistribution Mechanism

The marine environment is complex and variable. AUVs encounter complications during cooperative operation tasks, including entangled propellers and collisions with seaweed. These events cause AUV failure and prevent the Multi-AUV system from achieving its preset mission objectives. It is important to consider the environment's effects when utilizing AUVs. Therefore, the Multi-AUV system requires a task reassignment capability to address AUV failures during the mission, which can result in mission failure. To tackle

this challenge, the DE-GWO algorithm, known for its faster problem-solving capabilities, was employed for task reassignment, as depicted in Figure 3.

During the initial assignment of multiple tasks, the Multi-AUV system can assign intricate collaborative tasks to the AUV. If the AUV malfunctions during the mission implementation and is unable to carry out the mission as initially planned, the Multi-AUV system sends out an alarm message to notify the researchers. Meanwhile, the Pilot within the system can redistribute tasks centrally after considering the completion status of the current collaborative tasks and the state of the remaining AUVs. If the Pilot AUV experiences a failure, the system will choose another AUV with higher remaining energy and a shorter distance from the center of the current AUV to take over as the new Pilot. Tasks will then be reassigned to decrease the time necessary for the new Pilot to finish its mission. The overall goal is to improve the quality of communication during the information interaction process and successfully accomplish remaining tasks.

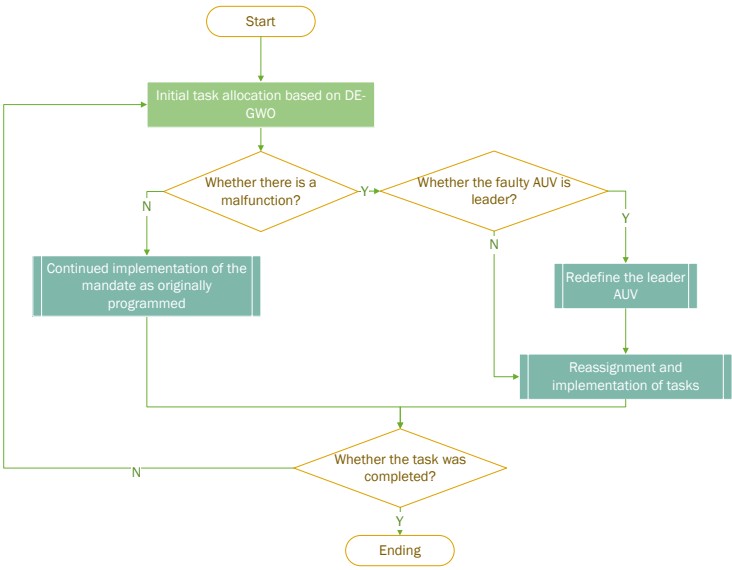

**Figure 3.** Task reassignment flowchart.

## 4. Results

In this study, simulation experiments were conducted using Matlab 2020 to assess the effectiveness and reliability of the DE-GWO algorithm in solving the Multi-AUV task-assignment problem. The paper presents a model of the marine environment for Multi-AUV cooperative operations, including the map of the environment and task points for the AUVs to complete before returning to the recovery point at the starting location. Given the expansive scope of the research objectives in the context of the vast oceanic area, the mission objectives were consolidated. Figure 4 shows the environmental maps.

In this paper, we classified tasks into three categories based on specific requirements. Each task requires the AUV to be equipped with the corresponding sensors before execution. In Table 2, we present information on the task configuration along with a representation graph in the result diagram. The starting point is symbolized by a blue dot, while the recovery point is depicted by a black pentagram.

The main objective of this paper was to facilitate the completion of ten target tasks by three AUVs and to determine the optimal Multi-AUV task-allocation approach while considering multiple constraints. In this paper, we conducted comparative experiments using the GWO algorithm, DE algorithm, PSO algorithm, and DE-GWO algorithm. We assumed that each AUV consumes one unit of energy when sailing a distance of 1 km, and a unit length in the environment map corresponds to a distance of 1 km in the actual marine environment. Thus, the energy consumption coefficient was one unit of energy per kilometer. Additionally, each AUV was equipped with different energy sources. The

experiment assumed that the three AUVs had an average speed of 2 km/h and it took each AUV 0.1 h to perform a task. The experiment's parameters specified that there were three AUVs, ten tasks, and a population size of 100 for the different algorithms, and each algorithm was executed in a loop 20 times.

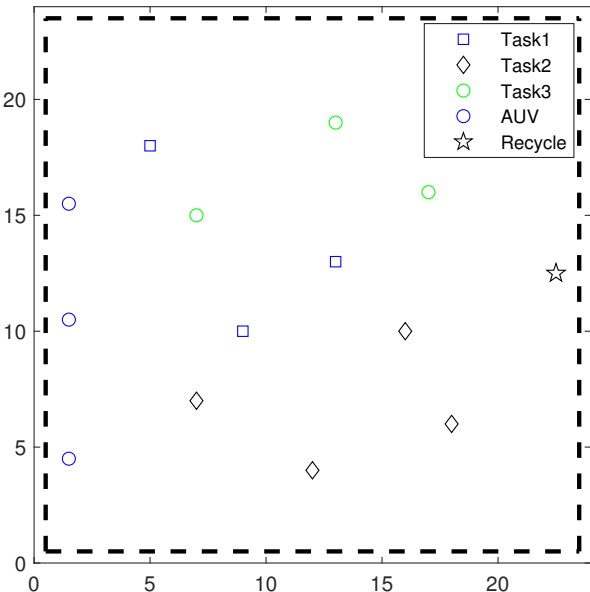

**Figure 4.** Mission environment map.

**Table 2.** Mission information sheet.

| Type of Task | Sensors Required for the Task | Task Number | Corresponding Graphs |
|---|---|---|---|
| *Task* 1 | Underwater camera | 2 5 6 | Green dots |
| *Task* 2 | Side-scan sonar | 3 4 9 10 | Black rhombus |
| *Task* 3 | Underwater camera and side-scan sonar | 1 7 8 | Blue square |

### 4.1. Multi-AUV Task Allocation with Equal Capability

In this section, we investigate the problem of task allocation for multiple AUVs with identical capabilities, but varying energy sources. We assumed that each AUV was equipped with the same sensors and was capable of performing all types of tasks. The three AUVs, each loaded with distinct energy sources, were deployed to fulfill ten operational tasks that had varying demands. It is essential to ensure that all constraints related to task allocation are met. The configuration details of the three AUVs can be found in Table 3.

**Table 3.** AUV information sheet.

| AUV Number | Number of Energy Sources | Sensor Type | Starting Point | Recovery Point |
|---|---|---|---|---|
| AUV 1 | 35 | Underwater camera and side-scan sonar | (1.5, 4.5) | (22.5, 12.5) |
| AUV 2 | 33 | Underwater camera and side-scan sonar | (1.5, 15.5) | (22.5, 12.5) |
| AUV 3 | 30 | Underwater camera and side-scan sonar | (1.5, 10.5) | (22.5, 12.5) |

Figure 5 illustrates the outcome achieved by assigning Multi-AUV systems of identical capacity and differing energy sources utilizing the GWO, DE, PSO, and DE-GWO algorithms, respectively.

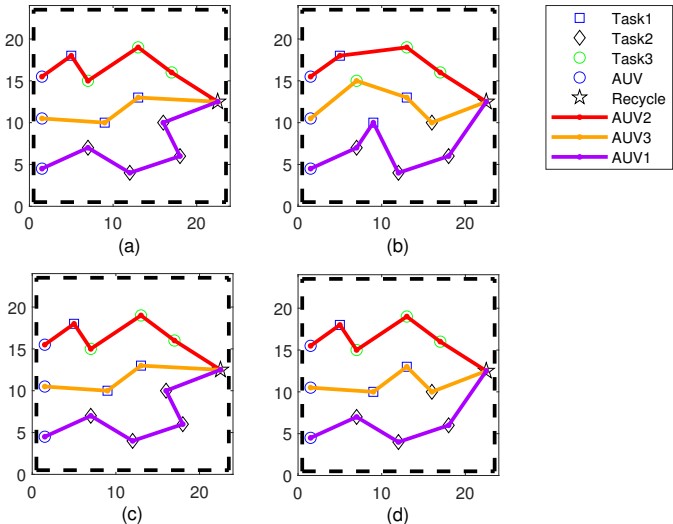

**Figure 5.** Comparison of task-assignment algorithms for a Multi-AUV system. (**a**) Results of task allocation using the GWO algorithm; (**b**) results of task allocation using the DE algorithm; (**c**) results of task allocation using the PSO algorithm; (**d**) results of task allocation using the DE-GWO algorithm.

When utilizing the GWO algorithm, the Multi-AUV system consumed 78.3002 of energy and took 15.2167 h to complete the task. In the case of the DE algorithm, the Multi-AUV system consumed 79.1058 of energy and took 15.6928 h. Similarly, with the PSO algorithm, the system consumed 78.3002 of energy and took 15.2167 h. Lastly, employing the DE-GWO algorithm resulted in the Multi-AUV system consuming 76.4632 of energy and taking 13.7185 h to complete the task. The detailed information about the resources consumed by each individual AUV is shown in Table 4.

**Table 4.** Multi-AUV system task-allocation strategy for different algorithms with AUVs with the same capability.

| Algorithm Name | AUV Number | Energy Consumption | Time Used to Complete Tasks | Sequence of Tasks Allocated |
|---|---|---|---|---|
| GWO | AUV 1 | 29.6334 | 15.2167 h | 3 4 9 10 |
| | AUV 2 | 26.6370 | 13.7185 h | 1 6 2 5 |
| | AUV 3 | 22.0298 | 11.2149 h | 8 7 |
| DE | AUV 1 | 30.5855 | 15.6928 h | 3 8 4 9 |
| | AUV 2 | 23.8826 | 12.2413 h | 1 2 5 |
| | AUV 3 | 24.6377 | 12.6189 h | 6 7 10 |
| PSO | AUV 1 | 29.6334 | 15.2167 h | 3 4 9 10 |
| | AUV 2 | 26.6370 | 13.7185 h | 1 6 2 5 |
| | AUV 3 | 22.0298 | 11.2149 h | 8 7 |
| DE-GWO | AUV 1 | 26.1027 | 13.3514 h | 3 4 9 |
| | AUV 2 | 26.6370 | 13.7185 h | 1 6 2 5 |
| | AUV 3 | 23.7235 | 12.1618 h | 8 7 10 |

When comparing different algorithms for the Multi-AUV task-allocation problem with equal capacity, it became evident that both the GWO and PSO algorithms produced the same optimal allocation results. Additionally, they slightly outperformed the DE algorithm in terms of energy and time consumption of the Multi-AUV system. Moreover, the DE-

GWO algorithm's optimal allocation results surpassed those of the other algorithms in terms of energy and time consumption. This algorithm also proved beneficial in reducing the task completion time and saving the energy consumed by the Multi-AUV system to complete the task.

Table 5 demonstrates that the DE-GWO algorithm outperformed other algorithms regarding the average objective function value and solving time in multiple solving processes. Additionally, this algorithm had a faster convergence speed and could discover a more-suitable task allocation scheme in a shorter period of time.

**Table 5.** Algorithm comparison table.

| Algorithm Name | Average Objective Function Value | Objective Function Variance | Algorithm Average Solution Time |
|:---:|:---:|:---:|:---:|
| GWO | 94.9813 | 1.71 | 7.52 s |
| DE | 95.1979 | 2.92 | 8.33 s |
| PSO | 95.4234 | 1.85 | 7.96 s |
| DE-PSO | 91.1158 | 1.53 | 6.12 s |

### 4.2. Multi-AUV Task Allocation with Different Capabilities

This section explores the issue of allocating tasks to the fleet of multiple AUVs with varied task execution capabilities and energy sources. The AUV possesses unique capabilities to execute tasks that should match the required target task specifications. The three AUVs, carrying different energy sources and equipped with a variety of sensors, must accomplish ten operational tasks, each with unique requirements, while meeting all mission assignment constraints. The configuration details of the three AUVs are displayed in Table 6.

**Table 6.** AUV information sheet.

| AUV Number | Number of Energy Sources | Sensor Type | Starting Point | Recovery Point |
|:---:|:---:|:---:|:---:|:---:|
| AUV 1 | 37 | Underwater camera | (1.5, 4.5) | (22.5, 12.5) |
| AUV 2 | 40 | Underwater camera and side-scan sonar | (1.5, 15.5) | (22.5, 12.5) |
| AUV3 | 35 | Side-scan sonar | (1.5, 10.5) | (22.5, 12.5) |

Figure 6 displays the various outcomes achieved by assigning the Multi-AUV system with diverse abilities and power sources utilizing the GWO, DE, PSO, and DE-GWO methodologies, respectively.

When utilizing the GWO algorithm, the overall energy consumption of the Multi-AUV system was 89.9397 and the task was completed within 15.4555 h. Conversely, if the DE algorithm was employed, the Multi-AUV system consumed a total energy of 92.3244 and operated for 18.4596 h. When using the PSO algorithm, the Multi-AUV system required 93.1948 of energy and 18.8948 h to complete the task. Finally, the DE-GWO algorithm consumed 86.1051 of energy and took 15.2494 h to complete the task when implemented. Table 7 presents a comprehensive summary of the individual AUVs' consumed resources.

Table 8 shows the comparison table of the DE-GWO algorithm with the GWO algorithm, DE algorithm, and PSO algorithm based on the Multi-AUV task-allocation problem with different capabilities and different energy sources.

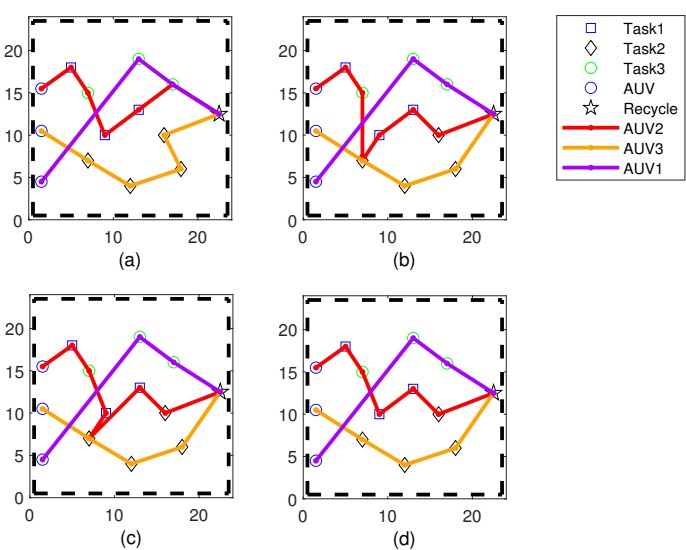

**Figure 6.** Comparison of task assignment algorithms for Multi-AUV system. (**a**) Results of task allocation using the GWO algorithm; (**b**) results of task allocation using the DE algorithm; (**c**) results of task allocation using the PSO algorithm; (**d**) results of task allocation using the DE-GWO algorithm.

**Table 7.** Multi-AUV system task-allocation strategy for different algorithms with the same capability AUV.

| Algorithm Name | AUV Number | Energy Consumption | Time Used to Complete Tasks | Sequence of Tasks Allocated |
|---|---|---|---|---|
| GWO | AUV 1 | 30.0176 | 15.1088 h | 2 |
| | AUV 2 | 29.8111 | 15.4056 h | 1 6 8 7 5 |
| | AUV 3 | 22.0298 | 15.4555 h | 3 4 9 10 |
| DE | AUV 1 | 30.026 | 15.213 h | 2 5 |
| | AUV 2 | 35.7191 | 18.4596 h | 1 6 3 8 7 10 |
| | AUV 3 | 26.5793 | 13.4897 h | 4 9 |
| PSO | AUV 1 | 30.0260 | 15.2130 h | 2 5 |
| | AUV 2 | 36.5895 | 18.8948 h | 1 6 8 3 7 10 |
| | AUV 3 | 26.5793 | 13.4897 h | 4 9 |
| DE-GWO | AUV 1 | 30.0260 | 15.2130 h | 2 5 |
| | AUV 2 | 29.4987 | 15.2494 h | 1 6 8 7 10 |
| | AUV 3 | 26.5804 | 13.5902 h | 3 4 9 |

In the study of task allocation for multiple AUVs with varying capabilities, the comparison of the results obtained from various algorithms demonstrated that the DE-GWO algorithm yielded superior outcomes regarding energy consumption and time consumption when compared to the other algorithms. According to Table 8, the DE-GWO algorithm can identify the superior task assignment approach in a shorter time, thereby enhancing the effectiveness of the Multi-AUV system.

Based on the data presented in Tables 5 and 8, distinct algorithms yielded differing outcomes for both the Multi-AUV task-allocation problem with similar capacities and those with varying capabilities. While the GWO algorithm can produce improved solutions, its global search proficiency was inferior and may result in a local optimal solution. Compared to the GWO algorithm, the DE-GWO algorithm exhibited superior global search abilities, faster convergence speeds, and greater algorithmic stability, enabling it to efficiently and effectively address the Multi-AUV task-allocation problem.

**Table 8.** Algorithm comparison table.

| Algorithm Name | Average Objective Function Value | Objective Function Variance | Algorithm Average Solution Time |
|---|---|---|---|
| GWO | 105.9953 | 2.52 | 11.76 s |
| DE | 111.1839 | 2.83 | 14.55 s |
| PSO | 112.9896 | 2.05 | 12.39 s |
| DE-PSO | 102.6181 | 1.77 | 9.34 s |

*4.3. Reassignment of Tasks*

Based on the Multi-AUV large-scale task-allocation experiment, we simulated the situation where one AUV fails and used the DE-GWO algorithm for task reassignment, assuming that each AUV had the same speed. AUV 2 failed at the "×" marking, as shown in Figure 7a. AUV 2 malfunctioned after completing Task 7, and Task 5 and Task 22 in its initial task sequence could not be completed as initially planned. The DE-GWO algorithm was used for task reassignment, which is shown in Figure 7b. After task reassignment, Task 5 and Task 22 were assigned to AUV 1, which was closer to the task point and had fewer initial tasks to execute.

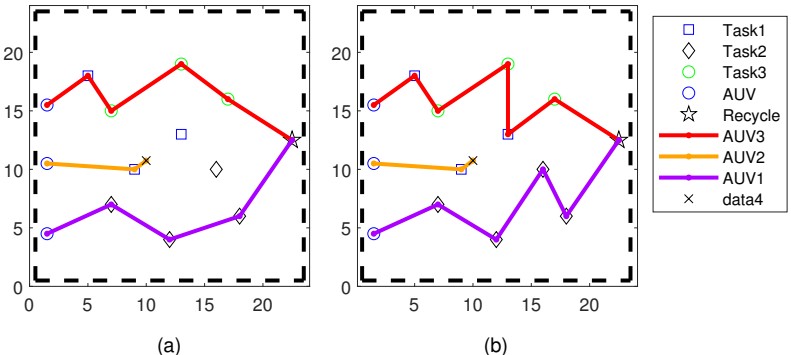

(a)   (b)

**Figure 7.** Comparison before and after task reassignment. (**a**) AUV 2 malfunctioned; (**b**) task reassignment results.

*4.4. Sea Trial Certificate*

In the process of multiple AUVs conducting cooperative operation, there are many disturbing factors that are difficult to reproduce in the simulation environment. In order to further verify the effectiveness of the method in the actual marine environment, three self-developed Model 260 AUVs were used to conduct the sea trial experiment.

Figure 8a illustrates the device model, which is the Model 260 AUV. This AUV boasts physical dimensions of 260 mm (outer diameter) × 2.5 m (length), weighs 100 kg in its standard configuration, and can achieve a maximum speed of 6–8 kn. In the context of the experiment, AUV 1, AUV 2, and AUV 3 represented the Model 260A AUV, Model 260B AUV, and Model 260C AUV, respectively.

The sea trial experiment area was situated in the offshore waters near Tundao Bay in Qingdao City, Shandong Province, China. A satellite map of the experiment area is depicted in Figure 8b, with red rectangles indicating the approximate boundaries of the experimental zone. The average water depth in the displayed sea area was approximately 10 m. To enhance clarity, the coordinates were converted and projected onto the geodetic coordinate system based on the Universal Transverse Mercator Grid System coordinates in meters. The sea trial experiment process is elucidated in Figure 8c,d.

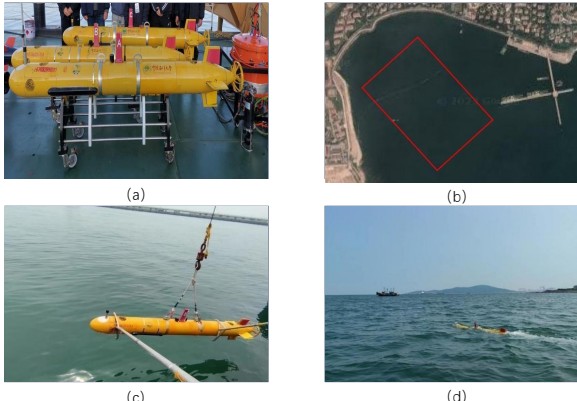

**Figure 8.** Sea trial experiment. (**a**) Static display of Type 260 multi-AUV system; (**b**) deployment of 260B; (**c**) sea trial experiment in progress; (**d**) AUV sailing in the water.

To validate the feasibility and effectiveness of the Multi-AUV task-allocation method proposed in this paper, the algorithm was implemented within the AUVs, and the designed task allocation method was executed on a physical platform. In the established experimental area, four locations were randomly chosen as the starting and recovery points for the three AUVs, while ten task points of various types were designated.

Assuming that the three AUVs possessed identical initial energy sources and consumed one unit of energy for each unit of path navigated, the experiment unfolded as follows: the three AUVs initiated their tasks from different starting points, completing their respective assignments before traveling to the recovery point. Detailed configuration information for the three AUVs can be found in Table 9.

**Table 9.** Algorithm comparison table.

| AUV Number | Equipped with Energy Source | Starting Point | Recovery Point |
|---|---|---|---|
| AUV 1 | 1500 | (31.53, 72.63) | (385.54, 202.33) |
| AUV 2 | 1500 | (30.96, 248.79) | (385.54, 202.33) |
| AUV 3 | 1500 | (32.26, 165.68) | (385.54, 202.33) |

In the Multi-AUV task-assignment experiment with identical capabilities, the three AUVs equipped with side-scan sonar and underwater cameras had the same operational capabilities. Three types of tasks were involved in this experiment, and these task types are detailed in Table 3. The task assignment results are presented in Table 10.

In the experiment involving Multi-AUV task assignment with varying capabilities, this section delimits the capabilities of the three AUVs. Specifically, the capabilities of AUV 1, AUV 2, and AUV 3 are outlined in Table 6. The results of the task assignments are displayed in Table 10.

**Table 10.** Results of Multi-AUV task assignment in the sea trial experiment.

| Status of AUV Capacity | AUV Number | Energy Consumption | Sequence of Allocated Tasks |
|---|---|---|---|
| Same-capacity AUV | AUV 1 | 576 | 3 4 9 |
| | AUV 2 | 599 | 1 6 2 5 |
| | AUV 3 | 522 | 8 7 10 |
| Different-capacity AUV | AUV 1 | 661 | 2 5 |
| | AUV 2 | 680 | 1 6 8 7 10 |
| | AUV 3 | 584 | 3 4 9 |

In conclusion, the task-allocation method based on the DE-GWO algorithm showed promising results in the real sea trial experiments. This approach effectively reduced the energy consumption of the Multi-AUV system, thereby enhancing its efficiency without compromising task completion. The experiments served as a validation of the feasibility of the DE-GWO algorithm.

## 5. Conclusions

In an effort to address the Multi-AUV task-allocation problem, this paper presented a DE-GWO algorithm and conducted an investigation. The initial step involved creating a mathematical model for the Multi-AUV task-allocation problem. Subsequently, the paper employed the GWO algorithm and incorporated Singer chaotic mapping for population initialization. Furthermore, the algorithm was enhanced by adopting a nonlinear convergence factor and dynamic weight updating strategy, thereby improving its ability to perform global searches and avoid local optima. The DE algorithm was then introduced, integrating the crossover, selection, and mutation operations into the optimization process. These operations continuously updated the population's position, enhanced its diversity, and accelerated the algorithm's convergence speed. Lastly, to assess the efficiency and effectiveness of the DE-GWO algorithm in solving the Multi-AUV task-allocation problem, several rounds of simulation experiments were conducted.

In the process of the research and summarization, this paper identified areas for potential improvement and enhancement, which are summarized in the following two points:

1.  While this paper introduced the DE-GWO algorithm, the complexity of Multi-AUV cooperative operation tasks necessitates higher real-time capabilities and flexibility in the task-allocation algorithm. Future research efforts should focus on studying more-dynamic task-allocation algorithms. This would enable the Multi-AUV system to dynamically adjust to environmental changes and real-time task modifications, allowing it to handle more-complex task situations with flexibility.
2.  The working state of AUVs during movement was not comprehensively considered, indicating a need for further improvement.

**Author Contributions:** Conceptualization, Z.C. and D.Z.; methodology, Z.C.; algorithm, Q.S. and Z.C.; validation, C.W. and D.Z.; investigation, Z.C. and C.W.; writing—original draft preparation, Z.C.; writing—review and editing, Z.C. and D.Z.; project administration, C.W. All authors have read and agreed to the published version of the manuscript.

**Funding:** This research received no external funding.

**Data Availability Statement:** The data presented in this study are available upon request from the corresponding author. The data are not publicly available because they are part of ongoing research and development.

**Acknowledgments:** The authors would like to thank Bo He, Jiaming Zhang, and Yunzhong Zhang for their technical support and thank the Editor and reviewers for their useful comments.

**Conflicts of Interest:** The authors declare no conflict of interest.

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
