# Peer review of "Hybrid Form of Differential Evolutionary and Gray Wolf Algorithm for Multi-AUV Task Allocation in Target Search"

_electronics, doi:10.3390/electronics12224575_

Round 1

Reviewer 1 Report

Comments and Suggestions for Authors

The article titled “Hybrid Form of Differential Evolutionary and Gray Wolf Algorithm for Multi-AUV Task Allocation in Target Search”, relates to my area of interest that’s why I recommend some points which may help in order to improve the readability as well as overall structure of this manuscript. The following are my suggestions, recommendations and questions for this article which may help to improve the quality of this manuscript are as follows:

1.     Abstract

·   Background of your proposed study may be properly elaborated  

·    By using the proposed novel method, how can you relate that your work is efficient than the other techniques.

2.    Introduction 

·       Background of the area must be elaborated more

·       Put some pictures of your relevant work   

·       Related work of the proposed algorithm may be elaborated more

·       Problem statement and solution are missing

·       Motivation and contribution part are missing

3.     Methodology 

·       Describe your proposed method "sufficiently" detailed such that others can redo your work.

·       Proposed work needs to be compared concisely with other related techniques to further discuss the effectiveness of your study.

4.     Discussions

·       Must include this part before conclusion

·       Define the potential application of the findings and limitations in this part

5.     References

     Check all the references carefully. The related reference written below may be cited and added in this section to further strengthen this work

General Comments

·       Background of the proposed area must be added.

·       Major observations in the introduction part as I reflect it on point number 3 so kindly incorporate all those points accordingly.

·       In methodology part, your proposed study must be sufficient enough so that others can redo your experiments. Also reflect effectiveness of your proposed work with proper details.

·       Discussion section must be added to highlight the overall progress and also mentioning the limitations of your proposed algorithm

·       Check all the references carefully and incorporate the above mentioned papers in your reference list.

Comments on the Quality of English Language

Extensive editing of English language required

Reviewer 2 Report

Comments and Suggestions for Authors

You need to improve the writing quality. There are many sentences with incomplete thought. And then there are sentences that serves no purpose.

Here are my concerns:

Line 5-6 (in Abstract): The sentence is incomplete. "Then, the advantages of Differential Evolutionary (DE) and Gray Wolf Optimization (GWO) Algorithm"....what? What do you plan to do with the advantages of DE and GWO?

Line 6 (in Abstract): Another confusion regarding your sentence: "Combining the advantages of DE algorithm"....with what? 

Line 10 (in Abstract): why didn't you compare the DE-GWO with DE as well? You just compared it to GWO and PSO. 

Lines 33 - 66: You have listed 1) swarm intelligent optimization, 2) SOM, and 3) market auction algorithms in your literature, but have only done a thorough review for SOM and market auction (including their deficiencies). The swarm intelligent algorithm was left unreviewed. Please make a short review for swarm intelligent algorithm and some of the problems it faces when applied to your task.

Line 90: List the problems with DE algorithm. If there are no problems with DE, then explain why you did not just use DE for your task, why you needed to combine DE with GWO?

Equation (5), you did not define "Goal"

Lines 202 - 206: what is the purpose of these sentences? Please remove.

Lines 244 - 246: It is not clear to me what you mean in this sentence.

Lines 373 - 378: Again, what is the purpose of these sentences? Please remove.

Figure 4: Please give legends. Reading forward, you do explain in Table 2 and in the paragraph below what the symbols in the figure refer to, but legends must still be there in the figure. blue circle = ?, green circle = ?, square = ?, diamond = ?, star = ?.

Line 460 - 461: The final sentence in this paragraph. The duration of which task? Is this an unnecessary sentence?

According to your results, it seems that DE is the worst performing algorithm in both scenarios, according to all three metrics: energy consumption, scenario completion time, and algorithm converging time. But in the review, you described DE in positive terms (compared to GWO, PSO). So I would expect the DE to perform better than PSO and GWO. How do you explain this difference between the expected and actual outcome?

Final note: in your scenario, how did you account for unexpected outcomes (like AUV getting damaged or stuck in see weed etc.). There was no scenario where task re-assignment had to take place. All 3 AUVs completed their assigned tasks. So you have not validated one critical part of your algorithm. You should show another scenario, where a random AUV randomly at a random point in time cannot carry on any further, and how your algorithm re-assigned the task of the damaged AUV to the other two AUVs).

Comments on the Quality of English Language

Here are my concerns:

Line 5-6 (in Abstract): The sentence is incomplete. "Then, the advantages of Differential Evolutionary (DE) and Gray Wolf Optimization (GWO) Algorithm"....what? What do you plan to do with the advantages of DE and GWO?

Line 6 (in Abstract): Another confusion regarding your sentence: "Combining the advantages of DE algorithm"....with what? 

Line 10 (in Abstract): why didn't you compare the DE-GWO with DE as well? You just compared it to GWO and PSO. 

Lines 33 - 66: You have listed 1) swarm intelligent optimization, 2) SOM, and 3) market auction algorithms in your literature, but have only done a thorough review for SOM and market auction (including their deficiencies). The swarm intelligent algorithm was left unreviewed. Please make a short review for swarm intelligent algorithm and some of the problems it faces when applied to your task.

Line 90: List the problems with DE algorithm. If there are no problems with DE, then explain why you did not just use DE for your task, why you needed to combine DE with GWO?

Equation (5), you did not define "Goal"

Lines 202 - 206: what is the purpose of these sentences? Please remove.

Lines 244 - 246: It is not clear to me what you mean in this sentence.

Lines 373 - 378: Again, what is the purpose of these sentences? Please remove.

Figure 4: Please give legends. Reading forward, you do explain in Table 2 and in the paragraph below what the symbols in the figure refer to, but legends must still be there in the figure. blue circle = ?, green circle = ?, square = ?, diamond = ?, star = ?.

Line 460 - 461: The final sentence in this paragraph. The duration of which task? Is this an unnecessary sentence?

According to your results, it seems that DE is the worst performing algorithm in both scenarios, according to all three metrics: energy consumption, scenario completion time, and algorithm converging time. But in the review, you described DE in positive terms (compared to GWO, PSO). So I would expect the DE to perform better than PSO and GWO. How do you explain this difference between the expected and actual outcome?

Final note: in your scenario, how did you account for unexpected outcomes (like AUV getting damaged or stuck in see weed etc.). There was no scenario where task re-assignment had to take place. All 3 AUVs completed their assigned tasks. So you have not validated one critical part of your algorithm. You should show another scenario, where a random AUV randomly at a random point in time cannot carry on any further, and how your algorithm re-assigned the task of the damaged AUV to the other two AUVs).

Reviewer 3 Report

Comments and Suggestions for Authors

This paper presents a new DEGWO algorithm to solve the multi-AUV task allocation problem.

The two great merits in this paper are:

1) The complete and clearly explained two-step process, first defining the mathematical model and then adapting bioinspired algorithms to solve it efficiently

2) The other great merit is is not to fall into the dark temptation of using connectionist architectures for the process. Bioinspired heuristics are more explainable and promote a better and complete tackling of the problems.

Also the proposed algorithm is compared with three other metaheuristics, and althoug results are very close, the proposed DEGWO algorithm slightly outperforms competitors.

Reviewer 4 Report

Comments and Suggestions for Authors

This paper aims to combine the global optimization capabilities of Differential Evolutionary (DE) with local optimization capabilities of Gray Wolf Optimization (GWO) to propose a DE-GWO algorithm for allocating tasks among multiple AUVs. Thus, combining the two algorithms offers a well-balanced solution to local and global optimization problems in multi-AUV task allocation.

Although the problem is well-defined but the quality of presentation and solution provided is not great.

I found a lot of spelling/grammar mistakes, such as:

Line 9: Task execution of Tasks

“In simulation comparison experimentsThe DE-GWO” to In simulations, comparison experiments between “The DE-GWO….” are carried out.

Line 11: no need for respectively

Lot of places, there is no space after “period”

Also, here are few concerns:

How do you distinguish Task in equation 3 and equation 4, where is definition of Goal, is the equation 4 represents the Goal?

Why the cost function J (equation 8) does not have a penalty on states (position and speed)? For example, you cannot move the AUVs at extreme speeds, also you can not move the AUVs backwards.

Line 333: How do you initialize the parameters?

Figures:

Figure 3: The text is going outside the boxes

Figure 4 and 5: There are some non-English characters present

Comments on the Quality of English Language

English language quality is poor

Round 2

Reviewer 2 Report

Comments and Suggestions for Authors

Minor issues:

A) Figures 5 & 6 are so compressed, it is hard to determine the icons. Either make larger figures, or increase the icon size (i.e., the size of the squares, circles, diamonds) in the figure so that they can be seen and understood clearly. I think the size of the figures in v1 of the report were more appropriate.

B) Figure 6 shows different result in v2 of the report compared to v1 of the report. This by itself is not concerning because these algorithms rely on random numbers and multiple simulations of the same scenario can result in different outcomes. What is surprising is that although Figure 6 in v2 of the report shows completely different outcomes compared to v1 of the report, Table 7 in v2 of the report still show the same sequences from v1 of the report. Thus, the contents of Table 7 in v2 of the report do not match the depiction of Figure 6 in v2 of the report.

C) Lines 450 - 452: the sentence is duplicated. Delete the repeating sentence.

Comments on the Quality of English Language

I have detected grammar problems regarding singular/plural. 

Reviewer 4 Report

Comments and Suggestions for Authors

I still feel that there should be a distinction between equation (3) and (4). Maybe (3) is like summation of Task. 
Also, if you do not impose a penalty on states then please explain how do you guarantee the state constraints are not violated?

Comments on the Quality of English Language

English language quality is fine

Round 3

Reviewer 4 Report

Comments and Suggestions for Authors

Authors have addressed my comments, hence, I am okay with this publication now. 

Comments on the Quality of English Language

English Quality needs to be inspected, I found some language errors.